# Microstructure and Properties of Mg-Al-Ca-Mn Alloy with High Ca/Al Ratio Fabricated by Hot Extrusion

**DOI:** 10.3390/ma14185230

**Published:** 2021-09-11

**Authors:** Aimin Chu, Yuping Zhao, Rafi ud-din, Hairong Hu, Qian Zhi, Zerui Wang

**Affiliations:** 1School of Materials Science and Engineering, Hunan University of Science and Technology, Xiangtan 411201, China; huhr@hnust.edu.cn (H.H.); 1200012@hnust.edu.cn (Q.Z.); 1920010221@mail.hnust.edu.cn (Z.W.); 2School of Civil and Engineering, Hunan University of Science and Technology, Xiangtan 411201, China; 1020090@hnust.edu.cn; 3Materials Division, PINSTECH, Post Office Nilore, Islamabad 44000, Pakistan; rafiuddi@gmail.com

**Keywords:** Mg alloys, high Ca/Al ratio, mechanical properties, hot extrusion

## Abstract

Mg-Al-Ca-Mn alloys with Ca/Al ≥ 1 of AX33, AX44, and AX55 were prepared by combining three processes of water-cooling semi-continuous cast, homogenization heat treatment, and hot extrusion. The as-fabricated alloys translated into composites consisting of α-Mg solid solution + granular Al_2_Ca. These alloys exhibited some favourable properties such as a tensile strength of 324~350 MPa at room temperature and 187~210 MPa at elevated temperature of 423 K, an ignition temperature of 1292~1344 K, and so on. Variation trend between performance and content of Al and Ca is given in this paper. The result indicated that the emerged second-phase Al_2_Ca in the alloys was beneficial to the improvement in mechanical properties, heat resistance, flame retardation, and corrosion resistance.

## 1. Introduction

In recent years, magnesium alloys have been attractively used in vehicle and aircraft industries due to their low density and high specific strength [1,2]. However, the low strength, low ignition temperature, and bad corrosion resistance of magnesium alloys restrain their widespread applications as structural materials [3,4,5]. The addition of Al to Mg alloy can form an Mg_17_Al_12_ phase to improve tensile strength [6]. The appending of Ca to Mg-Al alloy can form some second phases, improve mechanical properties [7], form a protective oxide layer containing CaO to improve flame retardation [8], and reduce the micro cell effect to improve corrosion resistance [9]. The addition of Mn to Mg alloy can refine grain [10]. Mg-Al-Ca alloys with Ca/Al <0.8 have been researched by many researchers in recent decades [11,12]. Nevertheless, little research about Mg alloy with Ca/Al ≥1 has been reported. To reveal the relationship between microstructure and properties of Mg alloy with high Ca/Al ratio, the alloys of Mg-3.2Al-3.3Ca-0.7Mn (AX33), Mg-4.2Al-4.3Ca-0.7Mn (AX44), and Mg-5.1Al-5.2Ca-0.7Mn (AX55) were fabricated and investigated in the present study.

## 2. Experimental Procedure

The alloys of AX33, AX44, and AX55 were melted by pure Mg, pure Al, pure Ca, and pure Mn and produced by water-cooling semi-continuous cast, homogenization heat treatment, and hot extrusion. The as-cast ingots were machined into a cylinder sample with the size of Ø 170 × 300 mm. After homogenization heat treatment at 673 K for 12 h, the cylinder samples were extruded into bars at 673 K with the extrusion ratio of 16. Test samples were cut parallel to extrusion direction or in the centre of the ingots.

Table 1 lists the chemical compositions of the alloys. The alloys possessed the same Ca/Al mass ratio of approximately 1. The addition of 0.7% Mn in the alloy contributed to removing Fe impurity and refining grain.

All polished specimens were etched using the etchant consisting of acetic acid (5 mL), alcohol (33 mL), picric acid (2 g), and deionized water (5 mL). Chemical analysis was conducted by induction-coupled plasma mass spectrometer (ICP). The microstructure was examined by scanning electron microscope (SEM) and optical microscope (OM). The phase studies of samples were performed by using XRD analysis (XRD, Rigaku, D8-Advance). Grain size was acquired by the linear intercept method, according to ASTM E112. Tensile tests were performed using samples with the gauge length of 23 mm and cross-sectional area of 6 mm × 2 mm by an Instron 3382 mechanical testing machine, according to ASTM B557, with strain rate of 1.5 × 10^−3^ s^−1^ at 423 K, 3.0 × 10^−3^ s^−1^ at 448 K, 3.6 × 10^−3^ s^−1^ at 623 K and 673 K, and 3.6 × 10^−4^ s^−1^ at 623 K and 673 K. Creep tests were carried out by using samples with the gauge length of 25 mm and cross-sectional area of 6 mm × 2 mm by a creep test machine (GWT304) at 423 K under 60~90 MPa. Ignition temperature tests were carried out by using the samples with the size of 20 mm × 20 mm × 20 mm by a resistance furnace test device in air. Tests of corrosion rate were carried out by using the samples with the size of 20 mm × 20 mm × 20 mm by weight loss corrosion measurement in 5% NaCl aqueous solution at 298 K. A potentiodynamic polarization test was operated by a CHI604C electrochemical analyzer and a three-electrodes’ system.

## 3. Results and Discussion

### 3.1. Results

It is clear in Table 2 that the ultimate tensile strengths (*σ*_b_) of the as-extruded alloys AX33, AX44, and AX55 were 324~350 MPa at room temperature and 187~210 MPa at elevated temperature of 423 K, respectively, creep total-elongation was 0.92~1.03% under 60 MPa at 448 K in 100 h, superplastic elongation-to-failure was 312~602% at 673 K under strain rate of 3.6 × 10^−4^ s^−1^, ignition temperature was 1292~1344 K in air, and corrosion rate was 1.3125~2.2492 g·mm^−2^·h^−1^ in 5% NaCl aqueous solution at 298 K. The yield strength (*σ*_s_) of the AX44 alloy was the highest among the three alloys. Among the three alloys, with the increasing of the content of Al and Ca in the alloy, the tensile strength at room temperature and elevated temperature increased first and then decreased, the creep total-elongation decreased, and the superplastic elongation-to-failure, ignition temperature, and corrosion rate increased. Compared with the AX33 and AX55 alloys, the AX44 alloy exhibited the best comprehensive balance performance for possessing the highest tensile strength, moderation ductility, and corrosion resistance.

### 3.2. Discussion

#### 3.2.1. Microstructure Characterization

Figure 1 shows XRD patterns of the as-extruded AX33, AX44, and AX55 alloys. It is obvious in Figure 1 that the three alloys consisted of α-Mg solid solution (α-Mgss) + Al_2_Ca phase. However, the formed type of the second phase in Mg-Al-Ca alloy was affected by Ca/Al mass ratio. For Ca/Al < 0.8, only the Al_2_Ca phase was formed. When Ca/Al > 0.8, (Mg,Al)_2_Ca or (Mg,Al)_2_Ca + Mg_2_Ca phases were formed [13,14]. During the high-temperature solidification process, (Mg,Al)_2_Ca phase first formed [14]. To some extent, the residual Al and Ca elements were also dissolved in the α-Mg solid solution. The microstructure of the molten metal formed at high temperature can be reserved by fast cooling [15]. The water-cooling semi-continuous cast with the cooling rate of approximately 373 K/s made the (Mg,Al)_2_Ca phase reserved in the as-cast ingots. Nevertheless, (Mg,Al)_2_Ca phase transformed into α-Mg solid solution and Al_2_Ca phase when homogenization treatment temperature was over 573 K [16]. Therefore, homogenization heat treatment made (Mg,Al)_2_Ca phase in the material samples turn into Al_2_Ca phase and α-Mg solid solution. There was no Mn or its compounds in the XRD pattern. The reason may be the following factors: (1) During the melting process, some Mn reacted with the impurity element Fe to form AlFeMn compound which precipitated to form slag; (2) some Mn dissolved into α-Mg matrix to form magnesium solid solution; (3) a trace of Mn might have formed Al-Mn compound, but its amount was too little to be examined by XRD. In addition, there were no low-melting-point phases of Mg_17_Al_12_ and Mg_2_Ca in the alloys. The reason is the Ca/Al ratio of approximately 1 made Al preferentially react with Ca, which led to inhibiting precipitation of Mg_17_Al_12_ phase, and the water-cooling semi-continuous cast made (Mg,Al)_2_Ca phase reserved, which gave rise to inhibiting precipitation of Mg_2_Ca phase.

Figure 2 displays OM graphs of the as-cast ingots of AX33, AX44 and AX55,. According to the documents of [13,14], when the calcium-to-aluminum ratio of magnesium alloy is greater than 0.8, the (Mg,Al)_2_Ca phase is formed after casting. Therefore,as can be concluded from Figure 2, the microstructure of the as-cast ingots consisted of reticular (Mg,Al)_2_Ca + α-Mgss. The reticular (Mg,Al)_2_Ca phase was distributed at the grain boundary of the α-Mg solid solution. The volume fraction of the (Mg,Al)_2_Ca phase in the as-cast ingots increased with as the content of Ca and Al increased. The average grain size of the AX33, AX44, and AX55 ingots was estimated as ~75 μm, ~55 μm, and ~45 μm, respectively.

Figure 3 shows SEM micrographs of the as-extruded alloys of AX33, AX44, and AX55. Combining SEM micrographs in Figure 3 and XRD results in Figure 1, it can be concluded from Figure 3 that the as-extruded alloys consisted of granular Al_2_Ca + α-Mgss. It is clear from Figure 3 that most of the Al_2_Ca particles with the size of approximately 2 μm uniformly dispersed at the grain boundary of α-Mgss, and the rest dispersed in α-Mgss. The volume fraction of the Al_2_Ca phase in the as-extruded alloy increased with the increasing content of Al and Ca. The extrusion process led to altering the distribution morphology of the Al_2_Ca phase and the refining grains. The Al_2_Ca particles were inclined to be distributed at the grain boundary of α-Mgss along the extrusion direction. The average grain size of the alloys of AX33, AX44, and AX55 was estimated as ~7.5 μm, ~5.5 μm and ~4.5 μm, respectively.

#### 3.2.2. Tensile Strength

It is suggested that the following factors affected room temperature strength of the as-extruded alloys. First, the extrusion process lowered the average grain size of the alloys from 45~75 μm to 4.5~7.5 μm. According to the Hall–Petch formula [12,17,18]:(1)$$\sigma =\sigma_{0}+k_{y}d^{-1/2}$$
where *σ* = strength, *σ*_0_ = constant, and *σ*_0_ about pure Mg is 90~120, *d* = average grain size, k_y_ = coefficient and k_y_ about Mg alloy is 280~320, strength of the as-extruded alloys can be markedly increased by grain refinement. Second, the high-strength Al_2_Ca particles in the alloys pin grain boundary of α-Mg solid solution and obstruct dislocation movement [19] will lead to dispersion strengthening. Finally, the elements Mn, Al and Ca to some extent dissolve in α-Mgss results in solid solution strengthening. On the basis of the Formula (1) and test value of tensile strength, by calculation, room temperature tensile strength can be added by grain refinement strengthening, dispersion strengthening and solid solution strengthening is approximately 100 MPa, 87 MPa and 45 MPa, respectively.

Also the three strengthening mechanisms play a role to elevated temperature strength at different extent. As far as dispersion strengthening is concerned, the influence of Al_2_Ca phase on the elevated temperature strength can be described as follows. When strength of the second phase is high, thermostability is good, content is moderate, morphology is spherically disperse, the size is fine, distribution is uniform, the elevated temperature strength will be improved by dispersion strengthening [20]. The Al_2_Ca phase is the only second phase in the as-extruded alloys. Compared with low melting point Mg_17_Al_12_ (*T*_m_ = 710 K, where *T*_m_ = melting point) and Mg_2_Ca (*T*_m_ = 987 K) phases, the second phase Al_2_Ca (*T*_m_ = 1352 K) has good thermostability. Extrusion process makes grains of Al_2_Ca phase refined and dispersed uniformly. During thermal deformation process at 423~448 K, the tiny, uniformly distributed, thermostable, high-strength Al_2_Ca particles in the alloy pin grain boundary and hamper dislocation climb leads to partly improving elevated temperature strength of the alloy [21].

As shown in Table 2, at room temperature and at elevated temperature of 423 K and 448 K, *σ*_b_ (AX33) < *σ*_b_ (AX44) > *σ*_b_ (AX55), that is, the tensile strength at elevated temperature and room temperature increased first and then decreased with the increasing content of Ca and Al in the alloy. It may depend on the volume fraction of Al_2_Ca phase in the alloy. The influence of Al_2_Ca phase on tensile strength of the as-extruded alloys has two aspects. On the one hand, it hampers dislocation movement of α-Mgss to improve strength of the alloy. On the other hand, overmuch of second phase in the alloy might cause stress concentration in grain boundary interface between Al_2_Ca phase and α-Mgss to decrease strength of the alloy. The increasing or decreasing of tensile strength at elevated temperature and room temperature depends on which aspect is the dominant. When the volume fraction of Al_2_Ca phase is about the optimum value, the as-extruded alloy owns the highest tensile strength.

#### 3.2.3. Creep and Superplasticity

During thermal deformation process, the relationship between strain rate (*ε*) and flow stress (*σ*) is given by the constitutive equation [22]:(2)$$\varepsilon =A\sigma^{n}\exp(-Q/RT)$$
where *T* = temperature, A = constant, *Q* = activation energy, *R* = gas constant, *n* = stress exponent. According to the Formula (2), the following formulas can be obtained:(3)$$n={\left[\frac{\partial \ln\varepsilon }{\partial \ln\sigma }\right]}_{Q,T}\approx -\frac{\Delta \ln\varepsilon }{\Delta \ln\sigma }=-\frac{\ln\varepsilon_{2}-\ln\varepsilon_{1}}{\ln\sigma_{2}-\ln\sigma_{1}}=\frac{\ln\varepsilon_{1}/\varepsilon_{2}}{\ln\sigma_{2}/\sigma_{1}}$$
(4)$$m=\frac{1}{n}=\frac{\ln\sigma_{1}/\sigma_{2}}{\ln\varepsilon_{2}/\varepsilon_{1}}$$
(5)$$Q=nR\left[\frac{\partial \ln\sigma }{\partial \left(\frac{1}{T}\right)}\right]\approx nR\frac{\Delta \ln\sigma }{\Delta \left(\frac{1}{T}\right)}=nR\frac{\ln\sigma_{2}/\sigma_{1}}{\frac{1}{T_{2}}-\frac{1}{T_{1}}}$$
where m = strain rate sensitivity index.

As shown in Figure 4 and Table 2, at 423 K under 60 MPa in 100 h, total-elongation (AX33) > total-elongation (AX44) > total-elongation (AX55), that is, the creep resistance enhances with the increasing content of Al and Ca in the alloy. At 423~448 K, the *n*-value of the as-extruded alloys of AX33, AX44 and AX55 calculated by the Formula (3) is 6.88, 6.92 and 7.03 respectively, indicating the main creep mechanism of the as-extruded alloys at 423~448 K is dislocation climb [23]. During creep process, Al_2_Ca particles pin grain boundary and obstruct dislocation climb in α-Mgss gives rise to improving creep resistance of the as-extruded alloys. A large amount and small size of Al_2_Ca particles leads to increasing ability of pinning and obstructing. Increasing content of Al_2_Ca particles and decreasing average grain size leads to enhancing creep resistance of the as-extruded alloys.

It is clear in Table 2 that superplastic deformation at 623 and 673 K is controlled by strain rate and temperature. When temperature = constant, the superplastic increases with strain rate increasing. When strain rate = constant, the superplastic increases with temperature increasing. Under strain rate of 3.6 × 10^−4^ and at 673 K, δ (AX33) < δ (AX44) < δ (AX55), where δ = elongation-to-failure, that is, superplastic deformation capacity at given strain rate and temperature enhances with the increasing content of Al and Ca in the alloy. At 623~673 K, the m-value of the as-extruded alloys of AX33, AX44 and AX55 calculated by Formula (4) is 0.32, 0.33 and 0.34 respectively, the *Q*-value of the as-extruded alloys of AX33, AX44 and AX55 calculated by Formula (5) is 161, 162 and 164 kJ/mol respectively. The m-value indicates the main superplastic deformation mechanism at given strain rate and temperature is grain boundary sliding [24]. The *Q*-value is greater than that of pure magnesium lattice diffusion activation energy (135 kJ/mol). It indicates that the sliding of grain boundary is controlled by lattice diffusion [25,26]. During superplastic deformation process, the tiny, high strength Al_2_Ca particles coordinated grain boundary sliding of α-Mgss, at the same time, the dynamic recrystallization of α-Mgss took place and the Al_2_Ca particles pinned grain boundary and inhibited the grain growth of α-Mgss. During superplastic deformation process, the thermostable high-strength Al_2_Ca particles are not easy to deform leads to requiring a higher activation energy of dislocation creep to continuously maintain the superplastic deformation process [26], so the *Q*-value greater than pure magnesium lattice diffusion activation energy. Increasing Al_2_Ca particles in the alloys leads to enhancing ability of coordinating and pinning and thus enhancing superplastic deformation capacity.

#### 3.2.4. Flame Retardation and Corrosion Resistance

As shown in Figure 5, ignition temperature of the AX55 alloy is much higher than that of AZ80 alloy. It is obvious in Table 2 that the alloys of AX33, AX44 and AX55 display a high ignition temperature, and *T* (AX33) < *T* (AX44) < *T* (AX55), where *T* = ignition temperature, that is, flame retardation increases with the increasing content of Al and Ca in the alloy. High temperature oxidation process made the alloys form oxidation products of CaO, MgO [27] and trace of MnAl_2_O_4_ [28] on surface of the alloys. The oxides of CaO, MgO and MnAl_2_O_4_ formed protective oxide layer. As can be seen in Figure 6 (right side), after heat preservation at 1223 K for 0.5 h, protective oxide layer generated on surface of the AX55 alloy and Al_2_Ca phase. The Pilling–Bedworth ratio [29] of CaO and MgO is 0.65 and 0.81 results in forming a loose protective oxide layer, while the existence of Al_2_Ca particles in it leads to increasing compactness of the protective oxide layer. As melting points of Al_2_Ca (1352 K), CaO (2843 K), MgO (3125 K) and MnAl_2_O_4_ (2403 K) are very high, the protective oxide layer is thermostable at high temperature. The thermostable compact protective oxide layer cut off the touch between oxygen and α-Mg solid solution led to improving ignition temperature. The more the Al_2_Ca particles in the alloy are, the more the compactness of the protective oxide layer is, the higher the ignition temperature of the alloy is. Among the three alloys, the AX55 alloy possesses the highest ignition temperature due to it has the most Al_2_Ca particles.

Under the same test conditions, corrosion rate of AZ31 alloy is 13.546 g·mm^−2^·h^−1^. Compared with the AZ31 alloy, the as-extruded alloys display a good corrosion resistance. As shown in Figure 7, polarization curve of AZ31 alloy is higher than that of the AX55 alloy. It indicates corrosion current density of AZ31 alloy is greater than that of the AX55 alloy. During immersion corrosion process, chemical reaction corrosion and galvanic corrosion were occurred. The α-Mg reacted with H_2_O to form Mg(OH)_2_ protection membrane [30]. With the Mg(OH)_2_ protection membrane dissolving in NaCl aqueous solution, the alloy is to be constantly corroded.

As shown in Figure 8, pitting corrosion occurs on surface of the AX55 alloy after immersion in 5% NaCl aqueous solution at 298 K for 1 h. It was aroused by the galvanic corrosion between α-Mg matrix used as anode and grain boundary interface used as cathode and the chemical reaction corrosion between α-Mg matrix and NaCl aqueous solution [31]. The following factors result in a good corrosion resistance. First, standard electrode potential of calcium and magnesium under aqueous solution at 298 K is −2.90 V and −2.37 V, respectively. The addition of Ca increased self corrosion potential and decreased self corrosion current density of the as-extruded alloys led to improving corrosion resistance. Second, Al_2_Ca phase can be used as physical barrier to decrease chemical reaction corrosion [32]. Finally, the element Fe is very harmful to corrosion resistance, for it has high standard electrode potential (−0.036 V, under aqueous solution at 298 K) and causes severe galvanic corrosion [33]. The addition of Mn can partly remove Fe impurity gives rise to improving corrosion resistance. As can be seen in Table 2, *ε*(AX33) < *ε*(AX44) < *ε*(AX55), where *ε* = corrosion rate, that is, the corrosion rate increases with the increasing content of Ca and Al in the alloy, and overmuch of Al and Ca in the alloy decreased corrosion resistance. It is possible that increasing Al_2_Ca will lead to accelerating galvanic corrosion between α-Mg used as anode and grain boundary of Al_2_Ca used as cathode. In the meantime, to some extent the chemical reaction corrosion between Ca and H_2_O is also existed in the alloy, increasing Ca will lead to accelerating chemical corrosion between Ca and H_2_O.

## 4. Conclusions

(1) High Ca/Al ratio and fast cooling led to inhibiting precipitation of the low melting point Mg_17_Al_12_ and Mg_2_Ca phases. Homogenization heat treatment resulted in precipitating Al_2_Ca phase. Hot extrusion gave rise to refining grains and forming Al_2_Ca particles. The as-fabricated alloys turn into composites composed of granular Al_2_Ca + α-Mgss. With the increasing content of Ca and Al in the alloys, the strength of the alloys increases first and then decreases, the creep resistance, superplastic and flame retardation are enhanced, while the corrosion resistance decreases.

(2) The AX33, AX44 and AX55 alloys exhibit good mechanical properties, heat resistance, flame retardation and corrosion resistance, including an ulitmate tensile strength of 324~350 MPa at room temperature and 187~210 MPa at 423 K, a creep total-elongation of 0.92~1.03% under 60 MPa at 448 K in 100 h, a superplastic elongation-to-failure of 312~602% at 673 K under strain rate of 3.6 × 10^−4^ s^−1^, an ignition temperature of 1292~1344 K and a corrosion rate of 1.3125~2.2492 g·mm^−2^·h^−1^ in 5% NaCl aqueous solution at 298 K.

(3) dispersion strengthening, grain refinement strengthening, and solid solution strengthening led to improve tensile strength. Under the experiment conditions, the main creep mechanism is dislocation climb, the main superplastic mechanism is grain boundary sliding controlled by lattice diffusion. The Al_2_Ca particles not only can obstruct dislocation climb to improve elevated temperature strength and enhance creep resistance, but also can coordinate grain boundary sliding to enhance superplastic. The formed protective oxide layer containing Al_2_Ca, MnAl_2_O_4_ and CaO resulted in high ignition temperature. Corrosion resistance can be improved by the role of Ca, Mn and Al_2_Ca, but overmuch of Ca and Al in the alloy results in decreasing corrosion resistance.

## Figures and Tables

**Figure 1 materials-14-05230-f001:**
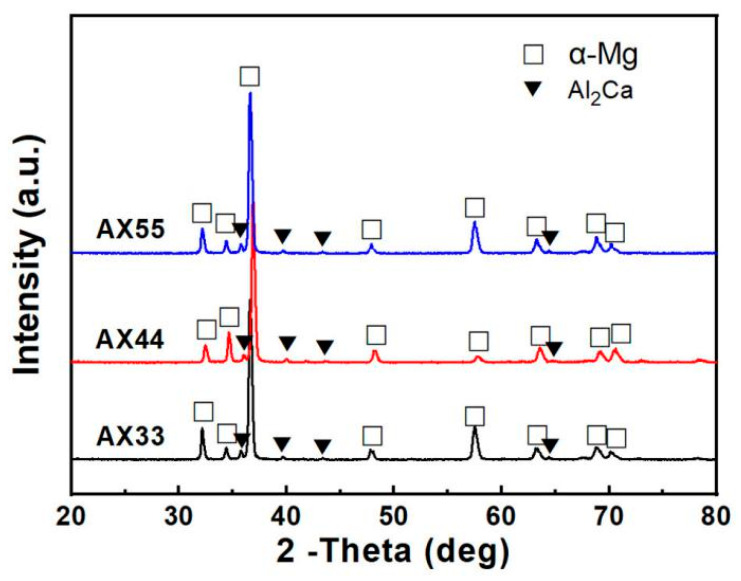
XRD patterns of the as-extruded AX33, AX44, and AX55 alloys.

**Figure 2 materials-14-05230-f002:**
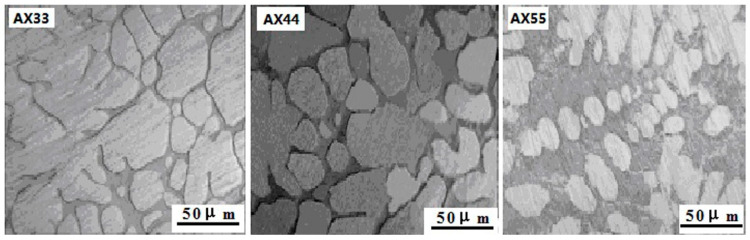
Optical micrographs of the as-cast ingots of AX33, AX44, and AX55.

**Figure 3 materials-14-05230-f003:**
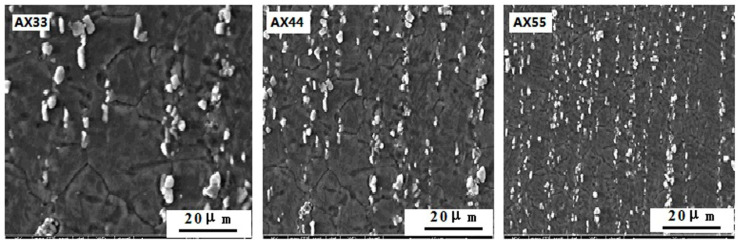
SEM micrographs of the as-extruded AX33, AX44, and AX55 alloys.

**Figure 4 materials-14-05230-f004:**
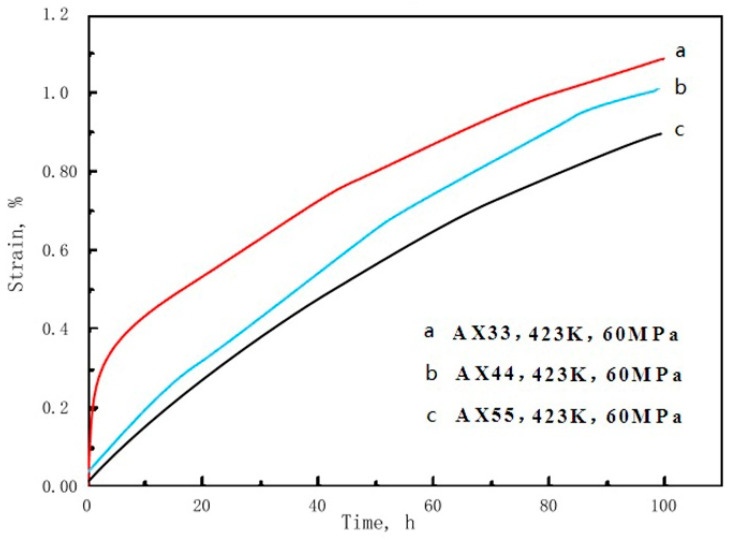
Creep curves of the AX33, AX44 and AX55 alloys at 423 K under 60 MPa in 100 h.

**Figure 5 materials-14-05230-f005:**
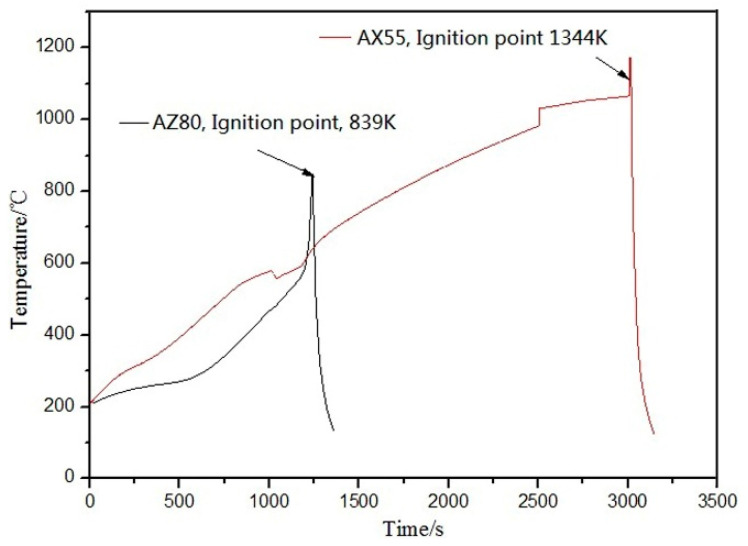
Ignition points of the AX55 alloy and AZ80 alloy.

**Figure 6 materials-14-05230-f006:**
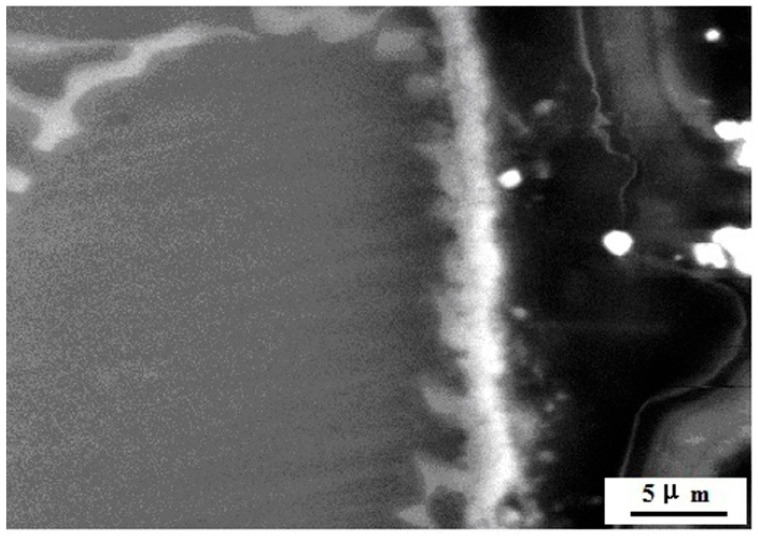
SEM morphology of cross-sectional oxide layer of the AX55 alloy after heat preservation at 1223 K for 0.5 h.

**Figure 7 materials-14-05230-f007:**
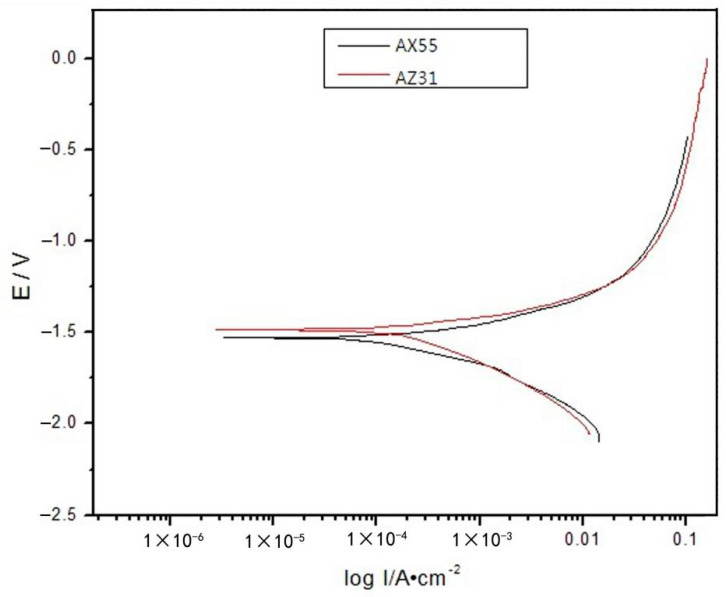
Polarization curves of the AX55 alloy and AZ31 alloy in 5% NaCl aqueous solution.

**Figure 8 materials-14-05230-f008:**
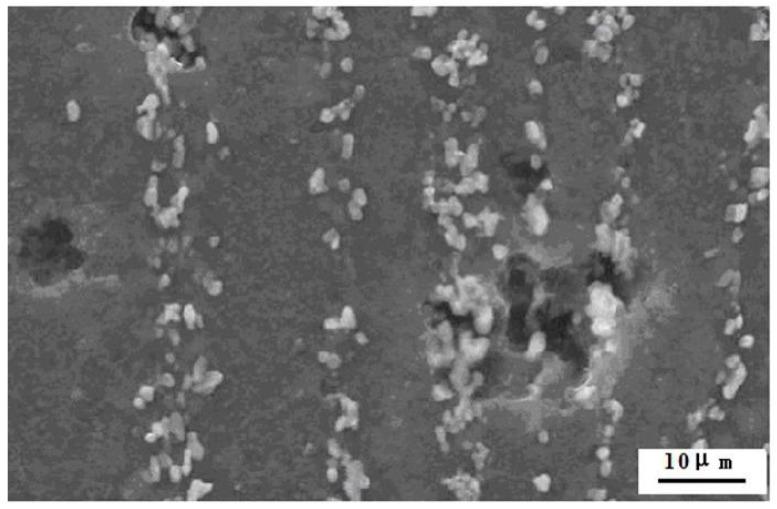
SEM morphology of the AX55 alloy after immersion in 5% NaCl aqueous solution at 298 K for 1 h.

**Table 1 materials-14-05230-t001:** Chemical compositions (wt%) of the AX33, AX44, and AX55 alloys.

	Al	Ca	Mn	Fe	Mg
AX33	3.20	3.30	0.70	<0.001	Bal.
AX44	4.20	4.30	0.70	<0.001	Bal.
AX55	5.10	5.20	0.70	<0.001	Bal.

**Table 2 materials-14-05230-t002:** Experiment conditions and properties of the AX33, AX44, and AX55 alloys.

	Experiment Conditions	Units	AX33	AX44	AX55
*σ*_b_/*σ*_s_/δ	RT	MPa/MPa/%	324/287/6	350/318/5	339/315/4
*σ*_b_/*σ*_s_/δ	423 K	MPa/MPa/%	201/150/16	210/153/14	187/152/13
*σ*_b_/*σ*_s_/δ	448 K	MPa/MPa/%	175/143/32	194/154/17	151/128/15
Creep total-elongation	423 K 60 MPa 100 h	%	1.03	0.98	0.92
Secondary creep rate	423 K 60 MPa 100 h	×10^−8^ s^−1^	2.05	1.95	1.98
	423 K 70 MPa 100 h	×10^−8^ s^−1^	3.04	2.89	2.68
	423 K 90 MPa 100 h	×10^−8^ s^−1^	32.6	27.2	26.5
superplasticity elongation-to-failure δ/*σ*_b_	673 K 3.6 × 10^−4^ s^−1^	%/MPa	312/4.3	384/8.3	602/9.5
	673 K 3.6 × 10^−3^ s^−1^	%/MPa	272/9.8	311/18.2	449/18.3
	623 K 3.6 × 10^−4^ s^−1^	%/MPa	213/17.9	263/20.3	352/20.5
	623 K 3.6 × 10^−3^ s^−1^	%/MPa	139/28.8	209/34.1	229/41.9
Ignition temperature	in air	k	1292	1313	1344
Corrosion rate	298 K in 5% NaCl aq	g·mm^−2^·h^−1^	1.3125	1.4861	2.2492

## Data Availability

The study did not report any data.

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
