# Peer review of "Microstructure and Properties of Mg-Al-Ca-Mn Alloy with High Ca/Al Ratio Fabricated by Hot Extrusion"

_materials, 2021, doi:10.3390/ma14185230_

Round 1

Reviewer 1 Report

In Table 2, the units of measurement for the corrosion rate should be converted to  g·m-2·h-1, then you will not need х10-6

It is not clear why in section 3.2.4. studies are carried out only with the AX55 alloy. Why is the AZ80 alloy used for comparison in some tests, and the AZ31 alloy in others? It is necessary to give an explanation of the choice.

Author Response

Reviewer 1

Comments and Suggestions for Authors:

1) In Table 2, the units of measurement for the corrosion rate should be converted to  g·m-2·h-1, then you will not need х10-6

The units in Table 2 and text have been modified according to respected referee’s advice in the revised manuscript now.

2) It is not clear why in section 3.2.4. studies are carried out only with the AX55 alloy. Why is the AZ80 alloy used for comparison in some tests, and the AZ31 alloy in others? It is necessary to give an explanation of the choice.

Considering the same phase composition of three alloys, therefore, the AX55 alloy was regard as a representative sample for discussing the flame retardant mechanism and corrosion mechanism. As a fact, both AZ80 alloy and AZ31 alloy are commercialized materials, however, two alloys always exist the problem of poor flame resistance and  corrosion resistance respectively. Therefore, in this paper, the main aim of using AZ80 alloy as comparison for flame resistance and AZ31 alloy as comparison for corrosion is to extend  the use of the AX55 alloy in the two fields.  

Thanks once again for your positive comments.

Reviewer 2 Report

Lines 59-60 Corrosion rate cannot be estimated by gravimetric method, as magnesium alloys are prone to pitting corrosion.
Lines 206 Why is the comparison with AZ80 alloy?
Lines 227 Why is the comparison with AZ31 alloy?
Lines 239 Figure 7:
1) absolutely identical graphs are shown - on what basis is the comparison of corrosion resistance?
2) why are logarithmic coordinates used?
Lines 247 Why is the potential of iron taken at -0.036V?

Author Response

Reviewer 2

1. Lines 59-60 Corrosion rate cannot be estimated by gravimetricmethod, as magnesium alloys are prone to pitting corrosion.

The advice of reverent reviewer is very reasonable. However, the gravimetric method is also an appropriate method for measuring corrosion rate of magnesium alloy. The corrosion of Mg alloy with high Ca/Al ratio acts by chemical corrosion and galvanic corrosion corrosion (As mentioned in section 3.2.4). The corrosion pit appearing in Figure 8 is a manifestation of galvanic corrosion of AX55 alloy.

2. Lines 206 Why is the comparison with AZ80 alloy?Lines 227 Why is the comparison with AZ31 alloy?
As a fact, both AZ80 alloy and AZ31 alloy are commercialized materials, however, two alloys always exist the problem of poor flame resistance and corrosion resistance respectively. Therefore, in this paper, the main aim of using AZ80 alloy as comparison for flame resistance and AZ31 alloy as comparison for corrosion is to extend the use of the AX55 alloy in the two fields.

3. Lines 239 Figure 7:
1) absolutely identical graphs are shown - on what basis is the comparison of corrosion resistance?

From Fig. 7, the self-corrosion potential of AZ31 (the tip intersection of the two curves) is higher than that of AX55.
2) why are logarithmic coordinates used?

Self-corrosion currents were calculated from the analysis software (chi760d).
3) Lines 247 Why is the potential of iron taken at -0.036V?

The electrode potential value of the iron (-0.036V) results from relevant document, and the document has been added in the revised manuscript now.

Thanks once again for your positive comments.

Reviewer 3 Report

The manuscript is devoted to study Microstructure and properties of Mg-Al-Ca-Mn alloy with high Ca/Al ratio fabricated by hot extrusion. It shows interesting experimental results however it is required to address the following comments.

1- in Table 1, what is σs?

2- It is recommended to show the XRD patternd of all materials for comparision

3- In Figures 2 and 3 how do you confirm the composition of (Mg,Al)2Ca and Al2Ca without EDS analysis. It is necessary to provide EDS results to prove the formation of these phases.

4- In Figures 1 and 3 you would better to add “extrusion” before AX…alloys.

5- Due to the difference in the grain size, Al2Ca size and content of Mn, Al,Ca solid solution effect, your calculated strengthening amounts belong to which materials?

Additionally, AX55 shows the smallest grain size and Al2Ca size and highest amount of solid solution elements. Means, theoretically it must show the highest strength while it does not match with experimental results. Some statement may be provided to explain this discrepancy?  You are also recommende to refer the following paper to explain the strengthnig mechanicms in the metallic materials. Ref: Deformation mechanism and enhanced properties of Cu–TiB2 composites evaluated by the in-situ tensile test and microstructure characterization

Author Response

Reviewer 3

The manuscript is devoted to study Microstructure and properties of Mg-Al-Ca-Mn alloy with high Ca/Al ratio fabricated by hot extrusion. It shows interesting experimental results however it is required to address the following comments.

1- in Table 1, what is σs?

The σs represents the yield strength of the three extruded alloys, and the illustration of σs  has been added in the revised manuscript now.

2- It is recommended to show the XRD patternd of all materials for comparision.

The XRD patterns of all materials for comparison have been added in the revised manuscript now.

3- In Figures 2 and 3 how do you confirm the composition of (Mg,Al)2Ca and Al2Ca without EDS analysis. It is necessary to provide EDS results to prove the formation of these phases.

The advice of reverent reviewer is very reasonable. Obviously, The EDS result can directly judge the phase composition in Figures 2 and 3. Unfortunately, our EDS results have been scheduled for publication in other papers. In fact, according to documents [13] and [14] in present paper, when the calcium-to-aluminum ratio of magnesium alloy is greater than 0.8, the (Mg,Al)2Ca phase is formed after casting, and the Al2Ca phase is formed after extruding according to the XRD diagram of these alloys.

4- In Figures 1 and 3 you would better to add “extrusion” before AX…alloys.

The term of extrusionhas been added in Figures 1 and 3 now.

5- Due to the difference in the grain size, Al2Ca size and content of Mn, Al,Ca solid solution effect, your calculated strengthening amounts belong to which materials?

Additionally, AX55 shows the smallest grain size and Al2Ca size and highest amount of solid solution elements. Means, theoretically it must show the highest strength while it does not match with experimental results. Some statement may be provided to explain this discrepancy? You are also recommended to refer the following paper to explain the strengthening mechanicms in the metallic materials. Ref: Deformation mechanism and enhanced properties of Cu-TiB2 composites evaluated by the in-situ tensile test and microstructure characterization.

The added strength value of three as-extruded alloys is calculated by comparing the strength of three casting alloys and that of three as-extruded alloys.

The effect of the Al2Ca phase content on the alloy strength comes from two aspects i.e. the second phase reinforcement and the crystal boundary stress concentration. Section 3.2.2 illustrates the effect of the Al2Ca phase content on the alloy strength. Moreover, the recommended document has been added to the reference catalogue now.

Thanks once again for your positive comments.

Round 2

Reviewer 1 Report

Сorrections accepted

Reviewer 3 Report

The authors have addressed my comments and I am satisfied with the improvements. Therefore, this manuscript is acceptable for publication in Materials.